# A Probability Contrastive Learning Framework for 3D Molecular Representation Learning

**Jiayu Qin**
University at Buffalo
jiayuqin@buffalo.edu

**Jian Chen**
University at Buffalo
jchen378@buffalo.edu

**Rohan Sharma**
University at Buffalo
rohanjag@buffalo.edu

**Jingchen Sun**
University at Buffalo
jsun39@buffalo.edu

**Changyou Chen**
University at Buffalo
changyou@buffalo.edu

## Abstract

Contrastive Learning (CL) plays a crucial role in molecular representation learning, enabling unsupervised learning from large scale unlabeled molecule datasets. It has inspired various applications in molecular property prediction and drug design. However, existing molecular representation learning methods often introduce potential false positive and false negative pairs through conventional graph augmentations like node masking and subgraph removal. The issue can lead to suboptimal performance when applying standard contrastive learning techniques to molecular datasets. To address the issue of false positive and negative pairs in molecular representation learning, we propose a novel probability-based contrastive learning (CL) framework. Unlike conventional methods, our approach introduces a learnable weight distribution via Bayesian modeling to automatically identify and mitigate false positive and negative pairs. This method is particularly effective because it dynamically adjusts to the data, improving the accuracy of the learned representations. Our model is learned by a stochastic expectation-maximization process, which optimizes the model by iteratively refining the probability estimates of sample weights and updating the model parameters. Experimental results indicate that our method outperforms existing approaches in 13 out of 15 molecular property prediction benchmarks in MoleculeNet dataset and 8 out of 12 benchmarks in the QM9 benchmark, achieving new state-of-the-art results on average.

## 1 Introduction

We investigate the problem of learning representations from molecules, a field known as molecular representation learning (MRL). MRL has gained significant attention due to its critical role in enabling learning from limited supervised data for applications such as molecular property prediction[29, 33, 5] and drug design [14, 20, 22]. Molecular representation learning involves creating models that can derive meaningful and generalizable representations of molecules, which can then be used to enhance various downstream applications. Among the most common methods in MRL is contrastive learning (CL), which leverages large-scale unlabeled molecular datasets to learn robust representations. CL works by contrasting different augmentations of the same molecule to ensure that the model learns to recognize the essential features of the molecule, thereby improving performance on tasks such as molecular property prediction and drug design.

With the success of contrastive learning methods in computer vision and multi-modality pretraining [7, 27], various contrastive learning approaches have been proposed for molecular representation learning. MolCLR[33] introduces a contrastive learning framework specifically for molecular representation

38th Conference on Neural Information Processing Systems (NeurIPS 2024).

learning. It employs atom masking and edge removal as data augmentations, which enhances the performance of Graph Neural Network (GNN) models on a variety of downstream molecular property prediction benchmarks. In contrast, GraphMVP[20] incorporates both 2D topology and 3D geometry during pre-training, though its downstream tasks primarily utilize 2D topology. These methods highlight different strategies for applying contrastive learning to molecular data, focusing on unique aspects of molecular structures to improve learning efficacy.

Although existing works have demonstrated the success of contrastive learning in molecular property predictions, they still face a significant drawback: the reliability of "positive" and "negative" labels in augmented molecule pairs. For example, MolCLR[33] uses augmentations like atom masking and edge removal, which can lead to false negative pairs when molecules with similar structures and chemical properties are labeled as negatives. Similarly, GraphMVP [20], which incorporates both 2D topology and 3D geometry, can also mislabel structurally similar augmented molecules as negatives due to its augmentation processes. These augmentations often remove parts of the molecular graph, such as nodes, edges, and subgraphs, resulting in potentially incorrect pairings. This issue is exacerbated by the large volume and extensive augmentations applied to molecular datasets, naturally leading to numerous falsely aligned pairs.

The fundamental problem lies in the random nature of these augmentations. Existing molecular contrastive learning methods assign hard positive and negatives to molecule pairs and do not account for the probabilistic relationships between molecules. Figure 3 provides an example of false positives and negatives resulting from graph augmentations in MolCL[33] ,where two distinct graph augmentations are applied to enhance two different molecules. The augmented molecule pair originating from the same molecule is categorized as positive, while other molecule pairs within the same batch are considered negative. However, as illustrated in the figure, the correct contrastive learning setup should consider molecules with structural similarities as positive pairs, even when they originates from different molecules. In contrast, the same molecule subjected to different augmentation methods may also be considered negative due to structural dissimilarities. Existing methods like MolCLR [33] fail to maintain this distinction, where augmented pairs from the same molecule are always treated as positive, while pairs from different molecules within the same batch are always treated as negative, regardless of their structural similarity. This mislabeling results in false positives and negatives, undermining the effectiveness of the contrastive learning process.

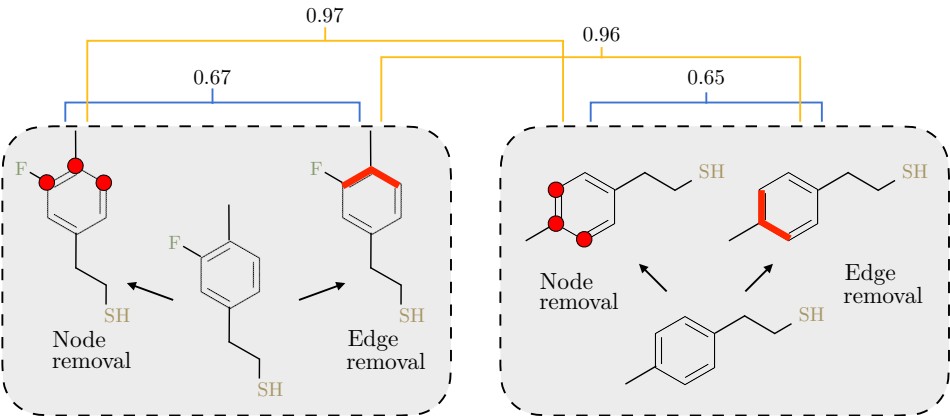

Figure 1: **Existing problem in molecular contrastive learning.** Adopt node removal and edge removal for molecular contrastive learning can lead to false positive and false negative problems. Blue lines indicate positive pairs and yellowing lines indicate negative pairs. The numbers on each line indicate the chemical similarity between the augmented pair of molecules. In this case, positive pairs indeed have lower similarity than negative pairs.

To overcome the aforementioned issue, we introduce a generalization of existing contrastive learning frameworks for molecular representation learning with probabilistic modeling. Our approach introduces data-pair weights as additional random variables, and dynamically infers optimal weights to account for false positive and false negative pairs, which can effectively address the mislabeling problem in previous methods. By incorporating a probability framework, we can effectively manage the uncertainty in data pair assignments. Specifically, we introduce a novel Bayesian inference

methods with Bayesian data augmentation to automatically infer these weights through posterior sampling. This allows us to optimize the model parameters efficiently using stochastic expectation maximization.

It is worth mentioning that while MolCLR[33] authors introduced i-MolCLR[32] to address similar issues by penalizing faulty negatives with a fingerprint-based similarity metric and a motif-level data augmentation called fragment contrast, our method offers distinct advantages. Unlike i-MolCLR which relies on direct fingerprint similarity, our approach introduces a novel probabilistic contrastive learning framework. This framework dynamically infers weight distributions and optimizes through stochastic expectation maximization, eliminating the need for explicit Tanimoto similarity calculations. Our method addresses the issue of false negative pairs more fundamentally and efficiently, providing a more robust solution for molecular contrastive learning.

In addition, our method is flexible and can be applied to different molecular representation learning framework. In this paper, we first integrate our method into MolCLR [33] series model and benchmark the performance on 2D non-charality MoleculeNet[35] dataset. We then integrated our method into Uni-Mol[42] and evaluate its performance on MoleculeNet[35]. We also trained and evaluated our model on the QM9 [28] dataset, following Equiformer [17]. With molecular property prediction tasks, we aim to test our model's ability in extracting useful features from molecular. Extensive experiments show that our method outperforms all other molecular representation learning baselines, including contrastive and non-contrastive methods.

The contributions of this paper can be summarized as follows:

- To tackle the challenges posed by false positive and negative pairs, we introduce a probability method for molecular contrastive learning. By introducing different weights as random variables to various false positive and negative pairs, we effectively mitigate the impact of these erroneous pairs on the learning process.
- To optimize our probabilistic contrastive learning framework, we propose a novel and effective optimization algorithm based on Bayesian data augmentation and stochastic expectation maximization, to simultaneously perform posterior inference and model optimization.
- Through extensive and large-scale experiments, we demonstrate enhanced performance across multiple public benchmarks for molecular representation learning, validating the effectiveness of our proposed method.

## 2 Methods

### 2.1 Learning Representations from Molecular Graphs

We begin by elucidating the foundational setup and notation in molecular contrastive learning. Molecules can be represented as 2D or 3D graphs depending on datasets. 2D molecule graphs have atoms as nodes and bond as edges. 3D molecule graphs additionally adds spacial positions of the atoms. For simplicity, we adopt static atom positions in this paper.

In molecular representation learning, as illustrated in Figure 2, we start by randomly sampling a batch of $N$ molecules. Each molecule, represented as $\mathbf{x}_i$, undergoes stochastic augmentation strategies to generate two augmented versions, denoted as $(\mathbf{x}_i, \mathbf{x}'_i)$. These augmentations involve methods such as atom masking, edge perturbation, and subgraph removal, transforming the original molecular structure while preserving its core characteristics. Among the resulting $2N$ augmented molecules, each pair $(\mathbf{x}_i, \mathbf{x}'_i)$ is treated as a positive pair, while the remaining $2(N-1)$ augmented molecules within the same batch are considered negative samples. This setup allows us to utilize contrastive learning effectively by distinguishing between similar and dissimilar molecular structures. A neural network encoder $f(\mathbf{x}; \boldsymbol{\theta})$, parameterized by $\theta$, is employed to extract representation vectors $z$ from the augmented molecular samples. In this paper, we utilize three different types of encoders in various experiments, as depicted in Figure 2 B, C, and D. These encoders include Graph Neural Networks (GNNs) and Transformers, each providing unique advantages for capturing the intricate features of molecular structures.

Let $s_{i+} \triangleq \mathrm{sim}(\mathbf{z}_i, \mathbf{z}'_i)$ represent the similarity score between the positive pair $(\mathbf{x}_i, \mathbf{x}'_i)$ after the encoder, and $s_{ik-} \triangleq \mathrm{sim}(\mathbf{z}_i, \mathbf{z}_k)$ signifies the similarity score between the negative pair $(\mathbf{x}_i, \mathbf{x}_k)$, and $\mathrm{sim}(\cdot, \cdot)$ represents any positive-valued similarity metric. In this paper, we adopt the commonly

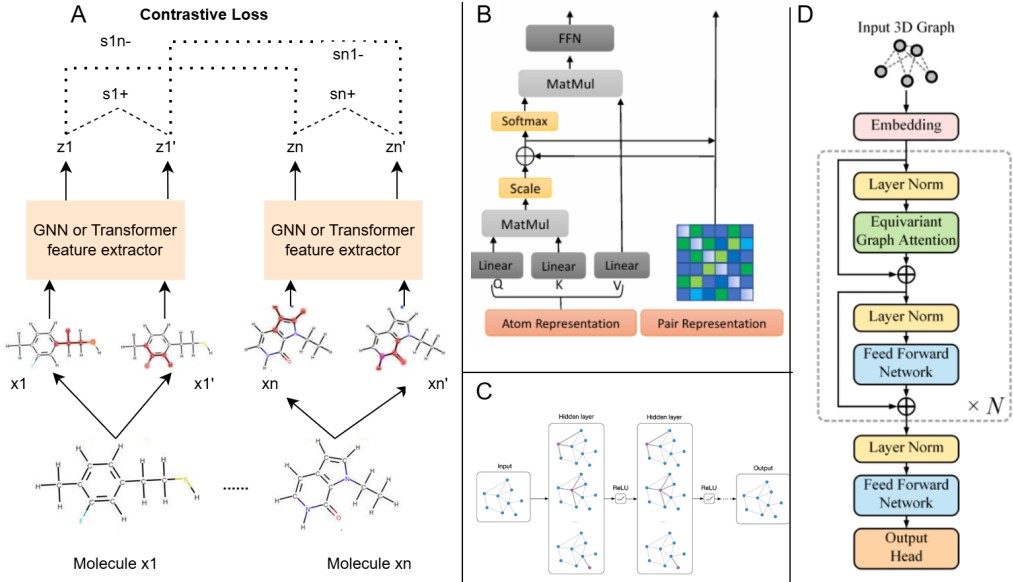

Figure 2: (A) **Molecular contrastive learning** Molecules are represented as 2D or 3D molecule graphs. Two stochastic augmentation strategies are applied to each graph, resulting in two augmentations. A feature extractor is used to extract features and contrastive loss is used to maximize the similarity of positive pairs and minimize the similarity of negative pairs B,C,D: Different architectures used as feature extractors in different experiments. (B) Uni-Mol [42] architecture used in MoleculeNet [35] Dataset experiment. (C) GCN [21] architecture from MolCLR [33] used in Non-Chirality MoleculeNet [35] experiment. (D) Equiformer [17] architecture used in QM9 [28] dataset experiment.

used exponential cosine similarity, defined as $\text{sim}(\mathbf{z}_1, \mathbf{z}_2) \triangleq e^{\mathbf{z}_1^T \mathbf{z}_2 / \|\mathbf{z}_1\| \|\mathbf{z}_2\| \tau}$, where $\tau$ denotes a temperature parameter.

## 2.2    Probability Weighted Contrastive Learning

We describe the proposed probability framework for molecular contrastive learning. In standard contrastive learning, one tries to encode data samples to a latent space such that positive pairs stay close to each other while negative pairs are pushed away. The contrastive loss function is:

$$\mathcal{L} = \frac{1}{N} \sum_{k=1}^{N} [\ell(2k-1, 2k) + \ell(2k, 2k-1)], \text{ with } \ell(i,j) = -\log \frac{s_{i+}}{s_{i+} + \sum_{k=1}^{2N} \mathbb{I}_{[k \neq i,j]} s_{i,k-}} \quad (1)$$

As mentioned, one issue of directly applying the contrastive learning into molecular representation learning is the potential false positive and negative molecular pairs, as discussed in the introduction. This could confuse the learning, ending up with sub-optimal representations. Is there a way to automatically identify and differentiate these pair data? In the following, we propose a Bayesian approach to address this issue that allows the algorithm for automatic inference of the degree of positiveness and negativeness of data pairs, involving enhancing the standard contrastive loss by incorporating learnable stochastic weights for all data pairs. To be more specific, we introduce local learnable weights, denoted as $w_i^+$ for each positive pair and $w_{ik}^-$ for each negative pair. We then define a weighted contrastive loss based on these introduced weights. This modification aims to mitigate the issues by automatically assigning relatively lower weights (or no weights) to false positive and false negative pairs;

$$\mathcal{L}_w = \frac{1}{N} \sum_{k=1}^{N} [\bar{\ell}(2k-1, 2k) + \bar{\ell}(2k, 2k-1)], \quad \bar{\ell}(i,j) = -\log \frac{w_i^+ s_{i+}}{w_i^+ s_{i+} + \sum_{k=1}^{2N} \mathbb{I}_{[k \neq i,j]} w_{ik}^- s_{ik-}} \quad (2)$$

One problem with this formulation, however, is that it is not realistic to compute and store all the weights in the learning process. This precaution arises from the quadratic growth in the number of

weights to be calculated as the training data size increases. Furthermore, the random nature of our augmentation method further adds complexity to the pre-calculation and storage of these weights.

A straightforward baseline for calculating these weights can be envisioned as follows: we can consider these weights in a binary fashion, with all weights initialized to one. In the learning process, if for some positive pairs the similarity score falls below a specified threshold, we set the corresponding weights to zero, marking these positive pairs as false positives. Conversely, if for some negative pairs the similarity score exceeds a threshold, we set the associated weights to zero, indicating false negatives. A challenge associated with this baseline method, however, lies in the establishment of a rigid similarity threshold to create a binary division of weights between zero and one. This approach proves less suitable for our molecular contrastive task as these heuristically chosen thresholds might not be optimal.

To address this challenge, we propose a principled Bayesian approach that allows adaptively inferring the optimal weights by Bayesian inference. Specifically, we treat the weights to be random variables and assign appropriate priors to them. We consider two types of priors: a Bernoulli prior to model weights as binary random variables and a Gamma prior to represent them as positive values. For simplicity, we model positive weights using the Gamma distribution and negative weights using either the Gamma distribution or the Bernoulli distribution, as expressed by the following formulas:

Option 1 - Gamma priors for continuous weighting:

$$w_i^+ \sim \text{Gamma}(a_+, b_+), w_{ik}^- \sim \text{Gamma}(a_-, b_-).$$

Option 2 - Bernoulli priors for selective weighting:

$$w_i^+ \sim \text{Gamma}(a_+, b_+), \quad w_{ik}^- \sim \text{Bernoulli}(a_-^-).$$

here, $a_+$, $b_+$, $a_-$ and $b_-$are shape and rate parameters for Gamma distribution and $a_-^-$ is the probability parameter for Bernoulli distribution.

With our reformulation, we can define a joint distribution over the global model parameter and local random weight variables $w_i^+$and $w_{ik}^-$, as:

$$p\left(\left\{w_i^+\right\}, \left\{w_{ik}^-\right\}, \boldsymbol{\theta}; \mathcal{D}\right) \propto \prod_{\mathbf{x}_i \in \mathcal{D}} \frac{w_i^+ s_{i+}}{w_i^+ s_{ij+} + \sum_{k=1}^K w_{ik}^- s_{ik-}} p(\{w_i^+\}) p(\{w_{ik}^-\}) p(\boldsymbol{\theta}). \quad (3)$$

One problem with the above formulation, however, is that posterior inference of the weights is challenging, due to the lack of convenience posterior distributions.

Fortunately, inspired by [2], we can introduce an augmented random variable $u_i$ that is associated to data point $\mathbf{x}_i$. Consequently, we can define an augmented joint posterior distribution of the random variables $\boldsymbol{\theta}, \mathbf{u}, \mathbf{w}$, denoted as $p\left(\left\{w_i^+\right\}, \left\{w_{ik}^-\right\}, \boldsymbol{\theta} \mid \mathcal{D}\right)$ [1], to be

$$p(\boldsymbol{\theta}, \mathbf{u}, \mathbf{w} \mid \mathcal{D}) \propto \prod_{i: \mathbf{x}_i \in \mathcal{D}} w_i^+ s_i + e^{-\mathbf{u}_i w_i^+ s_i+} \prod_k e^{-u_i w_{ik}^- s_{ik}-} p\left(\left\{w_i^+\right\}\right) p\left(\left\{w_{ik}^-\right\}\right) p(\boldsymbol{\theta}), \quad (4)$$

where $\mathbf{u} \triangleq \left\{u_1, u_2, \cdots, u_{|\mathcal{D}|}\right\}$ and $\mathbf{w} \triangleq \left\{w_i^+\right\} \cup \left\{w_{ik}^-\right\}$. It is worth noting that this joint distribution is equivalent to the original distribution (3), because (3) is recovered if one marginalize out the auxiliary random variables $\mathbf{u}$ in (4). In other words, optimization thought (4) is equivalent to optimization over (3). Consequently, we can perform learning and inference based on the augmented posterior of $p(\boldsymbol{\theta}, \mathbf{u}, \mathbf{w} \mid \mathcal{D})$, which preserves a much convenient form for posterior inference. In the following, we propose an efficient algorithm based on stochastic expectation maximization (stochastic EM) to alternatively infer the local random variables $\mathbf{w}$ and optimize the global model parameter $\boldsymbol{\theta}$.

## 2.3 Efficient Inference and Learning with Stocastic Expectation Maximization

We propose a stochastic EM algorithm for efficient inference and learning of our model. Stochastic EM [24] is a stochastic variant of the EM algorithm, which is an iterative method for finding the

---

[1]In the sense that marginalizing over the augmented random variables $\{w_i^+\}$and $\{w_{ik}^-\}$in $p\left(\theta, \mathbf{U}, \{w_i^+\}, \{w_{ik}^-\} \mid \mathcal{D}\right)$ gives back to the original $p\left(\{w_i^+\}, \{w_{ik}^-\}, \boldsymbol{\theta}; \mathcal{D}\right)$. Thus, learning and inferences on the two forms are equivalent.

maximum likelihood of model parameters in statistical models when data is only partially, or when model depends on unobserved latent variables [41].

In our setting, the objective of stocastic EM is to maximize the posterior in equation 4. The basic idea is to alternatively 1) optimizing model parameter $\boldsymbol{\theta}$ with fixed $(\mathbf{u}, \mathbf{w})$ and 2) sampling $(\mathbf{u}, \mathbf{w})$ with fixed $\boldsymbol{\theta}$. To this end, we follow standard procedures in stochastic EM to divide the learning into three steps: Simulation, Stochastic Expectation, and Maximization. Specifically, simulation corresponds to sampling local random variables $\mathbf{u}$ and $w$ for a batch of data; stochastic expectation then uses the sampled auxiliary random variables to update the model parameter $\boldsymbol{\theta}$ by maximizing a stochastic objective $Q(\boldsymbol{\theta})$, defined as: $Q_{t+1}(\boldsymbol{\theta}) = Q_t(\boldsymbol{\theta}) + \lambda_t \left( \log p(\boldsymbol{\theta}, \mathbf{u}, \mathbf{w} \mid \mathcal{D}) - Q_t(\boldsymbol{\theta}) \right)$ at iteration $t + 1$, where $\{\lambda_t\}$ is a sequence of decreasing weights. And maximization corresponds to maximizing the stochastic objective constructed in the previous step. In the following, we detail the three steps.

**Simulation** Given the joint posterior distribution in equation 3 and the current batch of data, the posterior distributions of the local random variables $\mathbf{u}$ and $\mathbf{w}$ can be directly read out, which simply follow Gamma or Bornoulli distributions of the following forms:

$$u_i \mid \left\{ w_i^+, w_{ik}^-, \boldsymbol{\theta} \right\} \sim \text{Gamma} \left( a_u, b_u + w_i^+ s_{i+} + \sum w_{ik}^- s_{ik-} \right), \forall i, \text{ and}$$

$$w_i^+ \mid \{\mathbf{u}, \boldsymbol{\theta}\} \sim \text{Gamma} \left( 1 + a_+, u_i s_{i+} + b_+ \right), \text{and}$$

$$\text{Option 1: } w_{ik}^- \mid \{\mathbf{u}, \boldsymbol{\theta}\} \sim \text{Gamma} \left( a_-, u_i s_{ik-} + b_- \right), \forall i, k \tag{5}$$

$$\text{Option 2: } w_{ik}^- \mid \{\mathbf{u}, \boldsymbol{\theta}\} \sim \text{Bernoulli} \left( \frac{a_- e^{-u_i s_{ik-}}}{1 - a_- + a_- e^{-u_i s_{ik-}}} \right)$$

We use the Gamma prior because it naturally lends itself to conjugacy in the posterior, which significantly eases the posterior sampling procedure. Also, it is known for its flexibility in shape and scale to model positive continuous variables, which is suitable for sample weights in our setting.

**Stochastic Expectation** We then proceed to calculate the stochastic expectation based on the simulated local random variables above. For notation simplicity, we define $Q_0(\boldsymbol{\theta}) = 0$. Then we can reformulate $Q_{t+1}(\boldsymbol{\theta})$ by decomposing the recursion, resulting in

$$Q_{t+1}(\boldsymbol{\theta}) = \sum_{\tau=0}^{t} \tilde{\lambda}_\tau \log p\left(\boldsymbol{\theta}, \mathbf{u}_\tau, \mathbf{w}_\tau \mid \mathcal{D}_\tau\right), \text{ where } \tilde{\lambda}_\tau \triangleq \lambda_\tau \prod_{t'=\tau+1}^{t} \left(1 - \lambda_{t'}\right), \tag{6}$$

where $\tau$ indexes the minibatch and the corresponding local random variables at the current time $\tau$.

**Maximization** The stochastic expectation objective (6) provides a convenient form for stochastic optimization over time, similar to online optimization (Bent & Van Hentenryck, 2005). Specifically, at each time $t$, we can initialize the parameter $\theta$ from the last step, and update it by stochastic gradient ascent on the log-likelihood, $\log p\left(\boldsymbol{\theta}, \mathbf{u}_\tau, \mathbf{w}_\tau \mid \mathcal{D}_\tau\right)$ calculated from the current batch of data. To reduce variance, we propose to optimize a marginal version by integrating out $\mathbf{u}_\tau$ from $p\left(\boldsymbol{\theta}, \mathbf{u}_\tau, \mathbf{w}_\tau \mid \mathcal{D}_\tau\right)$, which essentially reduces to our

---

**Algorithm 1** Contrastive Learning with Stochastic EM

---

1:  Initialize $\theta$; set $t = 1$
2:  **for** a batch of molecules in loader **do**
3:      Augment each molecule $\mathbf{x}_i$ into a pair $(\mathbf{x}_i, \mathbf{x}'_i)$
4:      Calculate positive/negative similarity scores $s^+$ and $s^-$ for all the molecule pairs
5:      Initialize all the weights $w^+$ and $w^-$ to be one
6:      **for** $k = 1$ to iter [4 in practice] **do**
7:          Sample u and w according to distributions
8:      **end for**
9:      Calculate the weighted contrastive loss in equation 2 with the sampled w on the current batch of data
10:     Update the model parameter by stochastic gradient descent with the calculated weighted contrastive loss
11:     $t = t + 1$
12: **end for**

---

original weighted contrastive loss in equation (2). With the above steps, it is ready to optimize the model by stochastic EM. The detailed steps are described in the Algorithm 1.

## 3   Related works

**Contrastive Learning** As a popular self-supervised learning paradigm, contrastive learning focuses on learning semantically informative representations for downstream tasks [16, 3, 39, 9]. The most widely used loss function is InfoNCE [25] which pulls in the representations between positive sample pairs while pushing away that between negative sample pairs.

**Molecular Representation Learning** Representation learning on large-scale unlabeled molecules attracts much attention recently. SMILES-BERT [31] is pretrained on SMILES strings of molecules using BERT. Subsequent works are mostly pretraining on 2D molecular topological graphs [15, 29]. MolCLR [33] applies data augmentation to molecular graphs at both node and graph levels, using a self-supervised contrastive learning strategy to learn molecular representations. I-MolCLR [32] is a improved version of MolCLR that uses new data augmentation and introduces weighted contrastive learning for mitigating false pair problem. Further, several recent works try to leverage the 3D spatial information of molecules, and focus on contrastive or transfer learning between 2D topology and 3D geometry of molecules. For example, GraphMVP [20] proposes a contrastive learning GNN-based framework between 2D topology and 3D geometry. GEM [5] uses bond angles and bond length as additional edge attributes to enhance 3D information. Uni-Mol[42] is a universal 3D molecular pretraining framework that significantly enlarges the representation ability and application scope in drug design.

**Noisy Pairs in Contrastive Learning** Noisy data pair problem have been found and studied in contrastive learning community. NLIP [11] enforces pairs with larger noise to be less similar in embedding space to improve model training. [6]apply noise estimation component to adjust the consistency between different modalities for action recognition task. RINCE [8] uses a ranked ordering of positive samples to improve InfoNCE loss. [3] introduces a new debiased contrastive learning loss function by transforming the distribution of negative samples. Matchdrop [34] designed a new graph augmentation method to alleviate the false positive sampling problem by retaining the most critical parts of the graph and augmenting the unimportant parts.

**Stochastic Expectation Maximization** Stochastic EM [24] stands as a pivotal algorithm in machine learning and probabilistic modeling for large-scale Bayesian inference. Building upon the foundations of the classical Expectation-Maximization (EM) algorithm [18], Stochastic EM offers an efficient solution for parameter estimation in situations involving vast datasets or latent variables, e.g., to maximize the $\log$-likelihood of $p(\mathbf{z}, \mathcal{D} \mid \boldsymbol{\theta})$, where $\mathcal{D}$ is the dataset, $\mathbf{z}$ is the local random variable and $\boldsymbol{\theta}$ is the global model parameter. By leveraging the power of mini-batch sampling, Stochastic EM strikes a balance between computational scalability and estimation accuracy. It has found widespread utility in various domains, including clustering [1], topic modeling [40], and latent variable modeling [41], making it an indispensable tool to cope with complex probabilistic models and extensive data and a natural fit to our problem.

# 4 Experiments

We evaluate our method on molecular property prediction tasks. Our approach is designed to be a versatile component that can be seamlessly integrated with various molecular property prediction datasets and models. In this study, we integrate our model into three different existing models: Uni-Mol[42]], I-MolCLR [32], Equiformer [17] and assess its performance on three distinct datasets: MoleculeNet [35], MoleculeNet without chirality, and the QM9 [28] dataset. For all experiments, we provide detailed experiment settings in Appendix C.

## 4.1 The MoleculeNet Dataset

MoleculeNet [35] is a popular benchmark for molecular property prediction, including datasets focusing on different molecular properties, from quantum mechanics and physical chemistry to biophysics and physiology. For a fair comparison, we integrated our method into Uni-Mol[42] framework. We applied both the *Gamma* and *Bernoulli* versions of our method, as shown in Table 1. In our contrastive learning framework, we used the representation of the [CLS] token as the final encoded representation, representing the entire molecule. Additionally, we incorporated the original three-dimensional recovery loss as an extra loss function. The model was trained on the same large-scale dataset, including 19 million molecules and 209 million conformations, as in the original paper. We used the same evaluation metrics: $ROC\_AUC$ for classification tasks and RMSE and MAE for regression tasks.

As shown in Table 1 and 2, our method outperforms Uni-Mol[42] and GEM [5], the current state-of-the-art methods, with an average gain of 1.3 percent in classification tasks and 7.6 percent in regression tasks. This substantiates that our approach facilitates more flexible training with a higher tolerance for false positive and false negative data pairs, thereby enhancing the model's performance in molecular representation learning.

Table 1: ROC_AUC on molecular property prediction classification tasks (Higher is better)

| Datasets | BBBP | BACE | ClinTox | Tox21 | ToxCast | SIDER | HIV | PCBA | MUV |
|---|---|---|---|---|---|---|---|---|---|
| # Molecules | 2039 | 1513 | 1478 | 7831 | 8575 | 1427 | 41127 | 437929 | 93078 |
| # Tasks | 1 | 1 | 2 | 12 | 617 | 27 | 1 | 128 | 17 |
| D-MPNN [37] | 71.0 | 80.9 | 90.6 | 75.9 | 65.5 | 57.0 | 77.1 | 86.2 | 78.6 |
| Attentive FP [36] | 64.3 | 78.4 | 84.7 | 76.1 | 63.7 | 60.6 | 75.7 | 80.1 | 76.6 |
| N-Gram$_{RF}$[19] | 69.7 | 77.9 | 77.5 | 74.3 | – | 66.8 | 77.2 | – | 76.9 |
| N-Gram$_{XGB}$[19] | 69.1 | 79.1 | 87.5 | 75.8 | – | 65.5 | 78.7 | – | 74.8 |
| PretrainGNN [10] | 68.7 | 84.5 | 72.6 | 78.1 | 65.7 | 62.7 | 79.9 | 86.0 | 81.3 |
| GraphMVP [20] | 72.4 | 81.2 | 79.1 | 75.9 | 63.1 | 63.9 | 77.0 | – | 77.7 |
| GEM [5] | 72.4 | 85.6 | 90.1 | 78.1 | 69.2 | **67.2** | 80.6 | 86.6 | 81.7 |
| MolCLR [33] | 72.2 | 82.4 | 91.2 | 75.0 | – | 58.9 | 78.1 | – | 79.6 |
| Uni-Mol[42] | 72.9 | 85.7 | **91.9** | 79.6 | 69.6 | 65.9 | 80.8 | 88.5 | 82.1 |
| Ours (Gamma) | **76.7** | **88.2** | 89.4 | **80.1** | **69.9** | 63.6 | **83.0** | **89.6** | 79.0 |
| Ours (Bernoulli) | 73.7 | 84.3 | 85.3 | 79.8 | 68.8 | 64.9 | 80.8 | 89.3 | **82.9** |

Table 2: Performance on molecular property prediction regression tasks (Lower is better)

| Datasets | ESOL | FreeSolv | Lipo | QM7 | QM8 | QM9 | MEAN (RMSE) | MEAN (MAE) |
|---|---|---|---|---|---|---|---|---|
| # Molecules | 1128 | 642 | 4200 | 6830 | 21786 | 133885 | | |
| # Metric | | RMSE↓ | | | MAE↓ | | | |
| D-MPNN [37] | 1.050 | 2.082 | 0.683 | 103.5 | 0.0190 | 0.00814 | 1.272 | 34.509 |
| GROVERlarge [29] | 0.895 | 2.272 | 0.823 | 92.0 | 0.0224 | 0.00986 | 1.33 | 30.67 |
| MolCLR [33] | 1.271 | 2.594 | 0.691 | 66.8 | 0.0178 | - | 1.519 | - |
| GraphMVP [20] | 1.029 | - | 0.681 | - | - | - | - | - |
| GEM [5] | 0.798 | 1.877 | 0.660 | 58.9 | 0.0171 | 0.00746 | 1.112 | 19.642 |
| Uni-Mol[42] | 0.788 | 1.480 | 0.603 | 41.8 | 0.0156 | 0.00467 | 0.957 | 13.940 |
| Ours (Gamma) | 0.775 | 1.420 | **0.590** | **38.5** | **0.0142** | **0.00395** | 0.928 | **12.839** |
| Ours (Bernoulli) | **0.664** | **1.358** | 0.626 | 55.6 | 0.0154 | 0.0056 | **0.883** | 18.541 |

## 4.2 Non-Chirality version MoleculeNet

In order to make a fair comparison with I-MolCLR [32], we also integrated our method into MolCLR [33] framework. MolCLR and I-MolCLR are 2D based methods, their experiments are conducted on different version of MoleculeNet dataset that does not consider chirality. We adopted the same dataset, augmentation, GNN-based encoder and other settings. As shown in Table 3, our method outperforms I-MolCLR on 7 out of 9 downstream tasks and got an average of 2 points increase on non-chirality MoleculeNet classification datasets.

Table 3: Comparison against i-MolCLR on non-chirality MoleculeNet dataset

| Without Chirality | BBBP | BACE | ClinTox | Tox21 | SIDER | HIV | MUV | MEAN |
|---|---|---|---|---|---|---|---|---|
| I-MOLCLR [32] | 76.4 | 88.5 | **95.4** | 79.9 | 69.9 | 80.8 | **90.8** | 83.1 |
| Our Method | **78.3** | **94.8** | 91.4 | **84.9** | **72.7** | **85.5** | 88.0 | **85.1** |

## 4.3 QM9 Dataset

The QM9 dataset [28] is another popular dataset in molecular property prediction, it consists of 134k small molecules, and the goal is to predict their quantum properties. For this dataset, we choose equiformer [17] as a baseline method. The data partition we use has 110k,10k,and 11k molecules in training, validation and testing sets. We use both our contrastive loss function and original minimize mean absolute error(MAE) as training objectives.

As shown in 4, we get state of the art result in 8 out of 12 baselines. The increase is relatively subtle compared with other dataset, we argue that this is due to the fact that QM9 is relatively small

regarding number of molecules in training set, and also the saturation on performance achieved by different methods.

Table 4: Experiment results on QM9 dataset

| Methods | $\alpha$ | $\Delta E$ | E_homo | E_lumo | $\mu$ | Cv | G | H | R^2 | $\mu$ | $\mu 0$ | ZPVE |
|---|---|---|---|---|---|---|---|---|---|---|---|---|
| GraphCL [39] | 0.066 | 45.5 | 26.8 | 22.9 | 0.027 | 0.028 | 10.2 | 9.6 | 0.095 | 9.7 | 9.6 | 1.42 |
| JOAOv2 [38] | 0.066 | 45.0 | 27.8 | 22.2 | 0.027 | 0.028 | 9.9 | 9.2 | 0.087 | 9.8 | 9.5 | 1.43 |
| 3D-MGP [12] | 0.057 | 37.1 | 21.3 | 18.2 | **0.020** | 0.026 | 9.3 | 8.7 | 0.092 | 8.6 | **8.6** | 1.38 |
| Transformer-M [21] | 0.041 | 27.4 | 17.5 | 16.2 | 0.037 | 0.022 | 9.63 | 9.39 | **0.075** | 9.41 | 9.37 | 1.18 |
| Equiformer [17] | 0.046 | 30 | **15** | 14 | 0.011 | 0.023 | 7.63 | 6.63 | 0.251 | 6.74 | 6.59 | 1.26 |
| Ours | **0.037** | **24.2** | 21.1 | **13.7** | 0.022 | **0.022** | **6.2** | **6.31** | 0.082 | **7.22** | 9.40 | **1.09** |

## 4.4 Ablation Study

**Distribution of similarity scores** Our method is largely motivated by the observation that previous MCL approaches neglect potential semantic dissimilarity between positive samples and that accounting for this phenomenon can improve learned molecule representations. In Figure A(See Appendix A), we plot the distribution of similarity scores for both positive and negative samples. Figure A left reveals that our method yields larger similarity scores with lower variance for positive pairs compared to MolCLR [33] baseline which uses standard contrastive learning method. Figure A right reveals that our method also mitigates the false negative problem in standard CL. It also shows that our method sometimes assigns lower similarity scores to positive pairs. While it may seem counter intuitive to assign lower similarity scores to positive samples, we argue that doing so is the very reason our method captures dissimilarity between positive pairs. By allowing some degree of alignment between the right set of negative examples, our method is able to minimize the inconsistencies between shared context of related positives and negatives. This in turn allows us to learn an overall more coherent representation space, resulting in increased robustness and downstream performance.

**Comparisons with the Standard Contrastive Learning** We conducted an ablation study to showcase that our method of probablistic framework of contrastive learning has already achieved strong emperical results and demonstrate the improvement brought by adding the 3D-aware loss functions on MoleculeNet [35] classification dataset. We first examined the effect of adding the probabilistic framework to the standard contrastive loss, and the 3D-aware loss functions as implemented in Uni-Mol[42].

Table 5: Ablation Study on MoleculeNet Classification Datasets

| | BBBP | BACE | ClinTox | Tox21 | ToxCast | SIDER | HIV | PCBA | MUV | MEAN |
|---|---|---|---|---|---|---|---|---|---|---|
| Standard CL | 69.3 | 81.5 | 84.1 | 75.5 | 63.4 | 58.9 | 78.3 | 84.1 | 72.5 | 75.2 |
| CL + 3D Loss | 75.1 | 86.8 | 87.9 | 78.9 | 68.5 | 62.8 | 81.8 | 88.0 | 77.1 | 78.1 |
| CL + Probabilistic Framework | 74.1 | 86.3 | 88.2 | 79.5 | 68.2 | 63.1 | 82.5 | 88.4 | 77.1 | 78.6 |
| CL + Both | **76.7** | **88.2** | **89.4** | **80.1** | **69.9** | **63.6** | **83.0** | **89.6** | **79.0** | **80.1** |

Table 5 presents the results of our ablation study. Incorporating the probabilistic framework resulted in a great improvement of 3.4-point increase in ROC-AUC, significantly enhances the model's performance. On the other hand, introducing the additional loss component led to an increase in ROC-AUC by 2.9 points, demonstrating its secondary role in enhancing the model's performance. When we adopt both of them, we can get the final ROC-AUC of 80.1 average on MoleculeNet classification datasets.

**Hyperparameters** We also conducted an ablation study to determine the optimal hyperparameters (e.g., $a_+$, $a_-$) on MoleculeNet classification datasets. We selected $a_+$, $a_-$, $b_+$, and $b_-$ from the range $[1, 5, 10]$. Table 6 indicates that our method achieves the best performance with $a_+ = 5$ and $a_- = b_+ = b_- = 1$. Tuning different hyperparameters affects performance, with an increase in $a_+$ from 1 to 5 leading to a 1.6 percent performance gain.

## 5 Conclusion

In this paper, we investigate an important yet unnoticed limitation of molecular contrastive learning, where augmented graph data come with false positive and false negative data pairs. As a remedy, we propose a principled solution to molecular contrastive learning by reformulating it into a probability

Table 6: Abalation studies on hyperparameters for MoleculeNet classification tasks

| $a_+$ | 1 | 5 | 10 | 5 | 5 | 5 | 5 |
|---|---|---|---|---|---|---|---|
| $a_-$ | 1 | 1 | 1 | 1 | 1 | 5 | 10 |
| b+ | 1 | 1 | 1 | 5 | 10 | 5 | 5 |
| $b_-$ | 1 | 1 | 1 | 1 | 1 | 5 | 10 |
| Avg. ROC-AUC (%) | 78.8 | **80.4** | 79.6 | 79.3 | 80.0 | 79.4 | 79.3 |

framework and introducing random weights for data pairs. With a Bayesian data augmentation technique, the random weights can be efficiently inferred via sampling, and the model parameter can be efficiently optimized via stochastic expectation maximization.

The effectiveness of our innovative approach has been proven through rigorous evaluations on multiple molecular property prediction benchmarks. The results also showcase the wide-ranging applicability and improved robustness of our proposed method over existing methods for learning molecular representations.

We believe our method is a valuable addition to the literature on molecular contrastive representation learning, which can further boost the performance of state-of-the-art molecular representation learning models for drug design.

## Acknowledgement

This work is partially supported by NSF AI Institute-2229873, NSF RI-2223292, NSF IIS-1747614 an Amazon research award, and an Adobe gift fund. Any opinions, findings and conclusions or recommendations expressed in this material are those of the author(s) and do not necessarily reflect the views of the National Science Foundation, the Institute of Education Sciences, or the U.S. Department of Education.

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

# A    Similarity Score Distribution

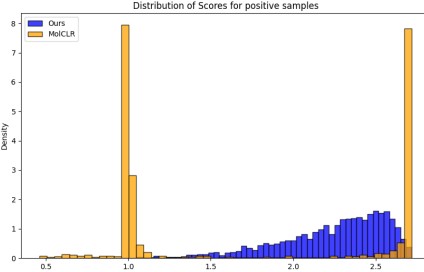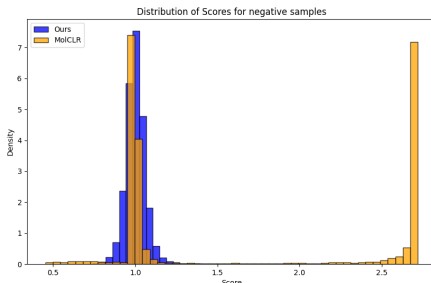

Figure 3: **Similarity Scores** – Similarity scores distribution for negative pairs in joint space after pre-training with original MolCLR loss and our proposed loss is provided. Compared to Using pretrained MolCLR model, our method yields similarity scores with lower mean and lower variance for negative pairs. While MolCLR have two peaks of negatives similarity scores around 1 and 2.7, our method concentrates them at only one peak of 1.Our method yields similarity scores with higher mean and lower variance for positive pairs. Our method concentrates at higher levels as it allows for some degree of semantic dissimilar between positives. The similarity scores are dot similarity, they are not normalized to enhance the difference for visual purposes.

# B    Limitations

In this section, we discuss the limitations of our proposed EM-based algorithm for molecular contrastive learning.

## B.1    Assumptions and Robustness

Our approach relies on several strong assumptions, such as the independence of molecular features and the noisiness nature of the input data. In practice, these assumptions may be violated, potentially affecting the performance and robustness of the model. For instance, correlated features could lead to biased estimates of weights, while unnoisy data might degrade the necessity to apply our method in learning representations. Future work could explore methods to relax these assumptions and enhance the model's robustness to such violations.

## B.2    Scope of Claims

The empirical results presented in this paper are based on experiments conducted on a specific set of datasets: MoleculeNet and QM9. While these datasets are commonly used in molecular machine learning research, they may not fully represent all possible application domains. Consequently, the generalizability of our findings to other datasets or real-world scenarios might be limited. Further validation on a broader range of datasets is necessary to confirm the wide applicability of our approach.

Also, one limitation of our method is that the performance gains brought by the proposed architectural improvements can depend on datasets and tasks. For small datasets like QM9, the performance gain is not significant.

## B.3    Privacy and Fairness

While our work does not specifically address issues of privacy and fairness, these are important considerations for any machine learning model, especially those used in sensitive domains such as healthcare. The potential for bias in molecular datasets, as well as privacy concerns related to molecular data, are areas that require further exploration. Ensuring that our model adheres to ethical standards and mitigates bias is an avenue for future work.

Table 7: hyperparameter search space for MoleculeNet dataset

| Hyperparameter | Small | Large | HIV |
|---|---|---|---|
| Learning rate | $[5e-5, 8e-5, 1e-4, 4e-4, 5e-4]$ | $[2e-5, 1e-4]$ | $[2e-5, 5e-5]$ |
| Batch size | $[32, 64, 128, 256]$ | $[128, 256]$ | $[128, 256]$ |
| Epochs | $[40, 60, 80, 100]$ | $[20, 40]$ | $[2, 5, 10]$ |
| Pooler dropout | $[0.0, 0.1, 0.2, 0.5]$ | $[0.0, 0.1]$ | $[0.0, 0.2]$ |
| Warmup ratio | $[0.0, 0.06, 0.1]$ | $[0.0, 0.06]$ | $[0.0, 0.1]$ |

## C   Training details for experiments

### C.1   MoleculeNet dataset

Following Unimol, we report the detailed hyperparameters setup of during pretraining in 7. Molecular pretraining runs on 4 A6000 GPUs, and the training time is about 48 hours. We split all the datasets with scaffold split, which splits molecules according to their molecular substructure.

### C.2   MoleculeNet Non-Chirality Analysis

We basically follow Mol-CLR on experiment settings. During pre-training, the GNN encoder maps each molecular graph to a 512-dimensional embedding $h$. A projection head, modeled as an MLP with a single hidden layer, transforms $h$ into a 256-dimensional latent vector $z$. ReLU is utilized as the non-linear activation function. The model undergoes pre-training over 50 epochs with a batch size of 512, optimized via the Adam optimizer with an initial learning rate of $5 \times 10^{-4}$ and a weight decay rate of $1 \times 10^{-5}$. A cosine learning rate decay schedule is applied throughout pre-training. For most datasets, we use a scaffold-based data split; however, the QM9 subtask follows a random split in line with the Mol-CLR methodology.

In the fine-tuning phase, the projection head is replaced with a randomly initialized MLP that maps $h$ to the target property prediction, while the pre-trained GNN encoder remains fixed. The fine-tuning process spans 100 epochs per benchmark task, with hyperparameters tuned via random search on validation sets, and results reported on test sets. Each benchmark is evaluated over three independent runs, with average performance reported. The model implementation is based on PyTorch Geometric.

### C.3   QM9 Dataset Experiments

Data partitioning for the QM9 tasks follows the scheme utilized in the Equiformer. For tasks predicting $\mu, \alpha, \varepsilon_{\text{HOMO}}, \varepsilon_{\text{LUMO}}, \Delta\varepsilon$, and $C_\nu$, our configuration includes a batch size of 64, 300 training epochs, a learning rate of $5 \times 10^{-4}$, and Gaussian radial basis functions with 128 bases. The architecture comprises six Transformer blocks, a weight decay of $5 \times 10^{-3}$, and a dropout rate of 0.2. Mixed-precision training is employed for these tasks.

For the $R^2$ task, the setup consists of a batch size of 48, 300 epochs, a learning rate of $1.5 \times 10^{-4}$, Gaussian radial basis functions with 128 bases, five Transformer blocks, a weight decay of $5 \times 10^{-3}$, and a dropout rate of 0.1, executed in single precision.

The ZPVE task employs a batch size of 48, 300 epochs, a learning rate of $1.5 \times 10^{-4}$, Gaussian radial basis functions with 128 bases, five Transformer blocks, a weight decay of $5 \times 10^{-3}$, and a dropout rate of 0.2, with single-precision training.

For the $G, H, U$, and $U_0$ tasks, the setup includes a batch size of 48, 300 epochs, a learning rate of $1.5 \times 10^{-4}$, Gaussian radial basis functions with 128 bases, five Transformer blocks, with both weight decay and dropout omitted, utilizing single precision.

All models were trained on a single A6000 GPU, with mixed-precision tasks requiring 81 GPU-hours and single-precision tasks requiring 151 GPU-hours. Model complexity includes 11.20 million parameters for configurations with six blocks and 9.35 million parameters for configurations with five blocks.

The data partitioning approach adheres to the random splitting strategy outlined in the Equiformer paper.

# D Comparison against other weight calculation methods

Here we show the comparison against using a simpler approach based on similarity scores. To thoroughly investigate this, we designed ablation experiments using the chiral version of MoleculeNet classification tasks and compared three different methods: 1. Bayesian Inference (Our Method): In this approach, we calculate the weights using Bayesian inference, as described in our paper. 2. Fingerprint-based Similarity: This method calculates the weights based on the similarity scores derived from molecular fingerprints, similar to the approach used in I-MolCLR [32]. 3. Encoder-based Similarity: Here, we first extract features of data pairs using encoders and then calculate their similarity scores. These scores are then regularized to the $[0, 1]$ range.

For methods 2 and 3 , we compute the weights using the following formulas:

$$w_i^- = 1 - \lambda \times \text{Sim}\left(x_i, x_k\right)$$
$$w_i^+ = \lambda \times \text{Sim}\left(x_i, x_k\right)$$

In our experiments, we set $(\lambda = 1)$.

| MoleculeNet Experiments | BBBP | Tox21 | ToxCast | SIDER | ClinTox | BACE | MUV | HIV | PCBA |
|---|---|---|---|---|---|---|---|---|---|
| Bayesian Inference | $76.7 \pm 2.0$ | $80.1 \pm \mathbf{1.0}$ | $69.9 \pm 2.5$ | $64.9 \pm 3.3$ | $89.4 \pm 0.1$ | $88.2 \pm 1.3$ | $82.9 \pm 3.1$ | $83.0 \pm 1.7$ | $82.9 \pm 0.5$ |
| Fingerprint Similarity | 77.9 | 80.0 | 68.9 | 64.9 | 87.6 | 83.6 | 78.7 | 79.8 | 83.8 |
| Encoder-based Similarity | 75.2 | 79.6 | 67.8 | 58.5 | 90.4 | 82.8 | 80.5 | 81.0 | 74.5 |

From these results, we observe the following:

1. While fingerprint-based similarity showed improvements in 2 out of 9 tasks compared to our original method, but it did not perform as well overall. This indicates that they may not be flexible enough to fully capture the complexities of molecular representations required for robust performance across diverse tasks.

2. Encoder-based Similarity performed worse than both the Bayesian inference method and the fingerprint-based similarity approach, further suggesting that using a direct similarity-based method does not necessarily yield better results.

These findings suggest that while simpler methods may work in some cases, they do not outperform our proposed Bayesian inference method which can dynamically adapt and provide better alignment of positive and negative pairs. Thus, our approach is essential for achieving state-of-the-art performance across various molecular property prediction benchmarks.

# E Protein-ligand binding task

We also conducted the protein-ligand binding pose prediction task. This is one of the most important tasks in structure based drug design. The task is to predict the complex structure of a protein binding site and a molecular ligand. We need to consider how ligand lays in the pocket, that is, the 6 degrees (3 rotations and 3 translations) of freedom of a rigid movement.

Following Uni-Mol, the molecular representation and pocket representation are firstly obtained from their own pretraining models by their own conformations; then, their representations are concatenated as the input of an additional 4-layer Transformer decoder, which is finetuned to learn the pair distances of all heavy atoms in molecule and pocket. Then, with the predicted pair-distance matrix as a scoring function, we first randomly place the ligand and then optimize the coordinates of its atoms by directly back-propagation the loss between current pair-distance and predicted pair-distance.

We evaluate our method using the metric binding pose accuracy. Specifically, we keep the pocket conformation fixed, while the ligand conformation is fully flexible. We evaluate the RMSD(root mean squared distance) between the prediction and the ground truth. Following previous works, we use the percentage of results below predefined RMSD thresholds as metrics.

The binding pose accuracy results are shown in Table. Not surprisingly, our model again outperforms all the baseline methods, achieving state-of-the-art results with our Gamma-prior version model.

Table 8: Performance on binding pose prediction.

| Methods | 1.0 Å | 1.5 Å | 2.0 Å | 3.0 Å | 5.0 Å |
|---|---|---|---|---|---|
| Autodock Vina[30, 4] | 44.21 | 57.54 | 64.56 | 73.68 | 84.56 |
| Vinardo [26] | 41.75 | 57.54 | 62.81 | 69.82 | 76.84 |
| Smina [13] | 47.37 | 59.65 | 65.26 | 74.39 | 82.11 |
| Autodock4 [23] | 21.75 | 31.58 | 35.44 | 47.02 | 64.56 |
| Uni-Mol[42] | 43.16 | 68.42 | 80.35 | 87.02 | 94.04 |
| Ours (Bernoulli) | **48.77** | **70.18** | 78.95 | 85.26 | 94.04 |
| Ours (Gamma) | 45.61 | 69.47 | **80.70** | **88.42** | **96.84** |

Table 9: ablation study on $a_u$ and $b_u$

| $a_u$ | 1 | 5 | 10 | 1 | 5 | 10 | 1 | 5 | 10 |
|---|---|---|---|---|---|---|---|---|---|
| $b_u$ | 1 | 1 | 1 | 5 | 5 | 5 | 10 | 10 | 10 |
| Avg ROC_AUC% | 80.4 | 80.2 | 80.1 | 80.6 | **80.7** | 80.2 | 80.3 | 80.1 | 80.5 |

## F   More ablation on $a_u$ and $b_u$

Here we conducted an ablation study on the choice of $a_u$ and $b_u$. Generally speaking, the choice of $a_u$ and $b_u$ will not influence the experiment results to a large margin, the top performance is at $a_u = b_u = 5$.

