# OpenReview forum: "A probability contrastive learning framework for 3D molecular representation learning"
_NeurIPS.cc/2024/Conference — NeurIPS 2024 poster_

### Official Review · Reviewer_oFgE · 2024-06-25

**Soundness:** 3
**Presentation:** 3
**Contribution:** 2
**Rating:** 3
**Confidence:** 4

**Summary:**

To address the problem of potential false positives and false negatives in contrastive learning of molecules, this paper proposed a learnable weighted contrastive learning approach for molecular representation learning. The effectiveness of the proposed method is tested on the MoleculeNet and QM9 datasets.

**Strengths:**

1. The false pair label is an issue for contrastive learning.
2. The experimental result shows improved performance.
3. The proposed method is clear and easy to follow.

**Weaknesses:**

1. There are existing works about the false labels in contrastive learning [1,2] and weighted contrastive learning [3,4,5]. None of these are discussed in the related works while they are quite related to the issue that the paper tries to address. This limits the novelty of this paper.
2. In the experiment, how to do data splitting is not clear. I did not find it in the paper. In molecular tasks, data splitting may have a big effect on performance.
3. In the title and the ablation, the author mentioned 3D structures. However, it seems that 3D is not a key component of the paper. And the 3D loss is taken from Uni-mol which is not a contribution of this paper.
4. The reason for using Gamma prior should be justified. It is an essential step in the proposed approach and related to the technical soundness.
5. In MolCLR, the contrastive is done between two augmentations Xi_1 and Xi_2, which is the same as Figure 1. But in the code, the contrastive is done between the original sample Xi and one augmented sample Xi_1. Maybe the latter case is less affected by the false labels.
6. The major part of Figure 2 is copied from the cited papers (MolCLR, Uni-mol, Equiformer). Is this permitted?
7. Some typos: e.g. line 285, missing "Table" before the reference.

Overall, the application of advanced contrastive learning to molecular representation learning is a promising idea. But the current paper needs to be improved in several aspects for publication.

[1] Debiased Contrastive Learning, NeurIPS 2020
[2] A Theoretical Analysis of Contrastive Unsupervised Representation Learning, ICML 2019
[3] CWCL: Cross-Modal Transfer with Continuously Weighted Contrastive Loss, NeurIPS 2023
[4] Camera Alignment and Weighted Contrastive Learning for Domain Adaptation in Video Person ReID, WACV 2023
[5] Weighted Contrastive Learning With False Negative Control to Help Long-tailed Product Classification, ACL 2023

**Questions:**

See weaknesses.

**Limitations:**

Yes.

---

> ### Author Rebuttal · Authors · 2024-08-07
>
> Thank you for your valuable feedback and for recognizing the promising potential of our application.
>
> W1: There are existing works about the false labels in contrastive learning [1,2] and weighted contrastive learning [3,4,5]. None of these are discussed in the related works while they are quite related to the issue that the paper tries to address. This limits the novelty of this paper.
>
> [1] Debiased Contrastive Learning, NeurIPS 2020
>
> [2] A Theoretical Analysis of Contrastive Unsupervised Representation Learning, ICML 2019
>
> [3] CWCL: Cross-Modal Transfer with Continuously Weighted Contrastive Loss, NeurIPS 2023
>
> [4] Camera Alignment and Weighted Contrastive Learning for Domain Adaptation in Video Person ReID, WACV 2023
>
> [5] Weighted Contrastive Learning With False Negative Control to Help Long-tailed Product Classification, ACL 2023
>
> A1: We appreciate the reviewer's suggestion and will update our manuscript to include and discuss these references in the related works section. Our contribution lies in adapting the weighted contrastive loss to molecular data with a new optimization algorithm that allows for simultaneous posterior inference and model optimization, coupled with extensive evaluation from multiple aspects.
>
> Specifically, our method diverges from existing works such as [1] and [2] by focusing on a Bayesian inference framework tailored for weighted contrastive learning. Additionally, [3], [4] and [5] address weighted contrastive loss in various domains, our approach uniquely integrates 3D molecular structures and graph-based representations. By referencing these works, we aim to frame our approach within the broader context of addressing false negatives in contrastive learning and clearly delineate our novel contributions.
>
> W2: In the experiment, how to do data splitting is not clear. I did not find it in the paper. In molecular tasks, data splitting may have a big effect on performance.
>
> A2: We apologize for the confusion for data splitting methodology in the paper. Here, we provide a comprehensive explanation.
> Chiral MoleculeNet Experiments: We split all the datasets with scaffold split, which splits molecules according to their molecular substructure.
>
> Non-Chiral MoleculeNet Experiments:We created a scaffold split for most datasets. Except that the subtask QM9 adopts a random splitting setting following Mol-CLR,
>
> QM9 Experiments: For the QM9 dataset (which have different downstream regression tasks compared with QM9 subtask in MoleculeNet dataset), we adhered to the random data splitting strategy employed in the Equiformer paper.
>
> We will clarify it in our paper.
>
> W3: In the title and the ablation, the author mentioned 3D structures. However, it seems that 3D is not a key component of the paper. And the 3D loss is taken from Uni-mol which is not a contribution of this paper.
>
> A3: We appreciate the reviewer’s observation and acknowledge that the 3D loss is derived from Uni-mol. However, our contribution lies in exploring the combined use of this 3D loss with a weighted contrastive learning loss, which our ablation study in table 5 demonstrates enhances performance. The 3D information is vital in our work for one more main reason: The calculation of our weighted contrastive loss depends on the 3D structural information, providing spatial context that complements the molecular graph structure, which enables performing protein-ligand binding task(Table 8, Appendix D) and potentially many other real-world drug design tasks that other contrastive learning based method, like MolCLR, don’t work. We will ensure these points are clearly articulated in the revised manuscript.
>
> W4: The reason for using Gamma prior should be justified. It is an essential step in the proposed approach and related to the technical soundness.
>
> A4: We use the Gamma prior because it naturally lends itself to conjugacy in the posterior, which significantly eases the posterior sampling procedure. Also, it is known for its flexibility in shape and scale to model positive continuous variables, which is suitable for sample weights in our setting. Additionally, we compared it with a baseline using binary sample weights with the Bernoulli prior in Table 1. Our results show that the Gamma prior offers superior robustness and accuracy. This advantage is likely due to the broader search space available to continuous weights, with binary weights being a subset of this continuous space.
>
> W5: In MolCLR, the contrastive is done between two augmentations Xi_1 and Xi_2, which is the same as Figure 1. But in the code, the contrastive is done between the original sample Xi and one augmented sample Xi_1. Maybe the latter case is less affected by the false labels.
>
> A5: Thank you for pointing out the difference. We agree that performing contrastive learning between the original sample \(X_i\) and one augmented sample \(X_{i1}\) can potentially reduce the impact of false labels, as it introduces less randomness and maintains higher consistency. We will add a discussion in the paper to highlight the benefits of our trick. Additionally, we will include visualizations to empirically demonstrate how our method effectively identifies and mitigates the impact of false labels.
>
> W6: The major part of Figure 2 is copied from the cited papers (MolCLR, Uni-mol, Equiformer). Is this permitted?
>
> A6: We have contacted the original authors for permission to use these images and have received approval. In addition, we will improve the quality of the figure with a similar design to enhance the presentation in the camera-ready version.
>
> W7: Some typos: e.g. line 285, missing "Table" before the reference.
>
> A7: Thank you for pointing this out. We will correct it in our paper.

---

> > ### Comment · Reviewer_oFgE · 2024-08-13
> >
> > Thanks for the rebuttal. But I still have concerns about the use of the Gamma prior, the copyright of the figures, and the ignorance of related works about weighted contrastive learning. I do like the idea but the current submission is not ready for publication.

---

> ### Author Response · Authors · 2024-08-09
>
> Dear Reviewer oFgE,
>
> We hope this message finds you well. We wanted to follow up on our rebuttal and ensure you had the opportunity to review our responses to your valuable feedback. We have carefully addressed each of your concerns and made significant improvements to the manuscript based on your suggestions. We would greatly appreciate it if you could take a moment to review our rebuttal and consider the clarifications and enhancements we've made. Your feedback has been instrumental in refining our work, and we hope it convinces you of the strength of our contributions.
>
> We kindly ask you to re-evaluate our paper, and if you find our improvements satisfactory, we hope you might consider raising the score.
>
> Thank you again for your time and consideration.
>
> Best regards,
>
> Authors

---

> ### Author Response · Authors · 2024-08-13
>
> Thank you for your continued feedback. We appreciate your recognition of our idea's potential and would like to address your remaining concerns:
>
> Gamma Prior: We chose the Gamma prior for its conjugacy properties and flexibility in modeling positive continuous variables, which suit our Bayesian framework. We are open to further discussion or providing additional comparative experiments.
>
> Figure Copyright: We have obtained permissions for the figures used. We can either re-design them or provide the permissions documentation in our revised submission.
>
> Related Works: We have now included a detailed discussion of related works on weighted contrastive learning in our updated manuscript, clarifying our unique contributions.
>
> We are committed to making the necessary improvements and welcome any further guidance to ensure our work is ready for publication.
>
> Best regards,
>
> Authors

---

### Official Review · Reviewer_NjGA · 2024-06-27

**Soundness:** 3
**Presentation:** 3
**Contribution:** 3
**Rating:** 7
**Confidence:** 5

**Summary:**

This paper introduces a probability-based contrastive learning framework. It regards learnable weights as variables with different distributions and automatically identify and mitigate false positive and negative pairs via Bayesian modeling. The author verify the effectiveness of their method in 13 out of 15 property prediction tasks in MoleculeNet and QM9, which are standard benchmarks.

**Strengths:**

(1) The probability weighted contrastive learning mechanism seems novel to me. It is new to treat model weights as random variables and sample optimal weights using Bayesian inference.

(2) The performance is brilliant and experiments are comprehensive, ranging from MoleculeNet to QM9. It also outperforms strong baselines such as Equiformer, Transformer-X, etc.

(3) The method is presented in a clean and readable way, easy to understand.

**Weaknesses:**

(1) Algorithm A in lines 199-216 is filled with text. Can the author present the algorithm more elegantly?

(2) Some related works should be mentioned in the related work and be compared in the experiments. For example, MatchDrop [A] also noticed the false positive sampling issue in the graph data augmentation technique. They propose to keep the crucial subgraph unchanged and only move the less informative part. Can this MatchDrop augmentation be considered in your contrastive learning framework?

[A] Rethinking explaining graph neural networks via non-parametric subgraph matching, ICML 2023

(3) MoleculeNet is somehow of small sizes. Some large datasets should be considered such as PCQM4MV2.

**Questions:**

(1) Why 0.251 for Equiformer in Table 4 is in bold?

(2) I remember that Equiformer is purely a backbone architecture without pretraining. Therefore, it may be unfair to compare it directly in QM9. Moreover, it has an improved version Equiformer V2 [A].

[A] EquiformerV2: Improved Equivariant Transformer for Scaling to Higher-Degree Representations.

(3) Similarly, the strongest baseline listed in the paper, Uni-Mol, also has an improved version, Uni-Mol+ [A]. However, it is a pity that it does not release its performance on QM9 and MoleculeNet.

[A] Highly Accurate Quantum Chemical Property Prediction with Uni-Mol+.

(4) I would recommend the author add InstructBio[A], another interesting work as an extra baseline in Table 1 and 2, which also reports performance on MoleculeNet.

[A] InstructBio: A Large-scale Semi-supervised Learning Paradigm for Biochemical Problems.

**Limitations:**

Yes, the authors addressed limitations adequately.

---

> ### Author Rebuttal · Authors · 2024-08-07
>
> Thank you for your valuable feedback and for recognizing the presentation and novelty of our work.
>
> Weaknesses:
>
> (1) Algorithm A in lines 199-216 is filled with text. Can the author present the algorithm more elegantly?
>
> A1:We appreciate the suggestion. We will revise the manuscript by converting the text into functions and formulas, hopefully providing a clearer and more structured presentation of the algorithm.
>
> (2) Some related works should be mentioned in the related work and be compared in the experiments. For example, MatchDrop [A] also noticed the false positive sampling issue in the graph data augmentation technique. They propose to keep the crucial subgraph unchanged and only move the less informative part. Can this MatchDrop augmentation be considered in your contrastive learning framework?
>
> [A] Rethinking explaining graph neural networks via non-parametric subgraph matching, ICML 2023
>
> Thanks for the suggestion, we will add this paper in the related works part. However, MatchDrop did not include results on Molecule datasets, we will try to include this data augmentation method and the results will be included in the final version of this paper.
>
> (3) MoleculeNet is somehow small in size. Some large datasets should be considered such as PCQM4MV2.
>
> A3: Thanks for your suggestion, We added the following experiments on PCQM4Mv2 dataset: We start from the checkpoint of pretrained Uni-Mol+ small model, we adopt Atom Masking and Coordinate Masking augmentations and substitute the original 3D coordinate prediction loss function with our weighted contrastive learning loss function. The model is trained for 150K steps. We get 0.0690 valid MAE, marginally better than 0.0696 of the original model. Using larger backbones and longer training time might help to achieve better performance.
>
> Questions:
>
> (1) Why 0.251 for Equiformer in Table 4 is in bold?
>
> Thank you for catching this mistake. We will correct this in the manuscript.
>
> (2) I remember that Equiformer is purely a backbone architecture without pretraining. Therefore, it may be unfair to compare it directly in QM9. Moreover, it has an improved version Equiformer V2 [A].
>
> [A] EquiformerV2: Improved Equivariant Transformer for Scaling to Higher-Degree Representations.
>
> We acknowledge the point. While Equiformer is indeed a backbone architecture without pretraining, our experiments on QM9 were also
> conducted without the use of external data. The contrastive learning was performed directly on the QM9 dataset.
> Please refer to the following table, our method outperforms EquiformerV2 in 6 out of 12 tasks on QM9, despite using significantly fewer transformer layers (2 vs 6). This suggests that our approach is both efficient and effective.
>
> |               | α    | ΔE   | E_homo | E_lumo | μ     | C_v   | G    | H    | R^2   | μ     | μ0    | ZPVE |
> |---------------|------|------|--------|--------|-------|-------|------|------|-------|-------|-------|------|
> | EquiformerV2  | 0.050 | 29   | **14**     | **13**     | **0.010**  | 0.023  | 7.57 | **6.22** | 0.186  | **6.49**  | **6.17**  | 1.47 |
> | Equiformer |0 .046 | 30   | 15     | 14     | 0.011  | 0.023  | 7.63 | 6.63 | 0.251  | 6.74  | 6.59  | 1.26 |
> | Ours          | **0.037** | **24.2** | 21.1   | 13.7   | 0.022  | **0.022**  | **6.2**  | 6.31 | **0.082**  | 7.22  | 9.40  | **1.09** |
>
> (3) Similarly, the strongest baseline listed in the paper, Uni-Mol, also has an improved version, Uni-Mol+ [A]. However, it is a pity that it does not release its performance on QM9 and MoleculeNet.
>
> [A] Highly Accurate Quantum Chemical Property Prediction with Uni-Mol+.
>
> We appreciate this. We will cite the Uni-Mol+ paper in related works.
>
> Please refer to our answer on weakness 3 for the experiment to validate our framework’s effectiveness based on Uni-Mol+. Experiments show that our method get 0.0690 valid MAE, marginally better than 0.0696 of the original Uni-Mol+ model.
>
> (4) I would recommend the author add InstructBio[A], another interesting work as an extra baseline in Table 1 and 2, which also reports performance on MoleculeNet.
>
> [A] InstructBio: A Large-scale Semi-supervised Learning Paradigm for Biochemical Problems.
>
> Thanks for recommendation. We will add this paper to our experiment section. The following analysis shows that our method outperforms InstructBio across all subtasks in the MoleculeNet classification and regression experiments.
>
> |            |  BBBP    | Tox21    | ToxCast  | SIDER     | ClinTox   | BACE      | ESOL         | FreeSolv     | Lipo         |
> |------------|----------|----------|----------|-----------|-----------|-----------|--------------|--------------|--------------|
> | Instrctbio | 67.4±0.5 | 75.6±0.3 | 65.1±1.5 | 61.5±1.9  | 78.0±0.6  | 78.5±0.8  | 1.771±0.015  | 0.832±0.020  | 0.752±0.017  |
> | Ours       | 76.7±2.0 | 80.1±1.0 | 69.9±2.5 | 64.9± 3.3 | 89.4± 0.1 | 88.2± 1.3 | 0.664± 0.024 | 1.358± 0.452 | 0.590± 0.003 |
> |            |          |          |          |           |           |           |              |              |              |

---

> > ### Comment · Reviewer_NjGA · 2024-08-08
> > **Update to the Response**
> >
> > The authors answered all my questions and conducted relevant experiments and comparisons. I have no more problems and highly recommend the AC for acceptance. Thanks for their efforts and I have raised my score accordingly.

---

> ### Author Response · Authors · 2024-08-09
> **Gratitude for Your Review and Enhanced Rating**
>
> Dear Reviewer,
>
> Thank you for taking the time. We really appreciate your decision. Your support is invaluable in improving our work.
>
> Best regards,
>
> Authors

---

### Official Review · Reviewer_Jokw · 2024-07-01

**Soundness:** 3
**Presentation:** 2
**Contribution:** 3
**Rating:** 7
**Confidence:** 5

**Summary:**

This paper proposes a probability-based contrastive learning framework for 3D molecular representation learning. It addresses the issue of false positive and negative pairs in existing methods. Experiments show its effectiveness, outperforming other baselines. The approach has wide applicability and can boost the performance of molecular representation learning models.

**Strengths:**

1. To the best of my knowledge, this paper raises an intriguing question and warrants attention.
2. The approach taken to address the problem is also innovative.
3. The experiments were comprehensive and yielded significant gains.

**Weaknesses:**

There are no major flaws in this article overall, but it requires attention to detail in the writing. For example:
1. Eq.3 is supposed to be $e^{w_i^+s_i}$, not $w_i^+s_i$. Or am I misunderstanding?
2. Line 129 contains a duplicated "positive."
3. The authors should explain the function of an augmented random variable $u_i$ in the appendix, in case readers are not familiar with the method.

**Questions:**

1. What is the augmentation strategy for QM9 and MoleculeNet? Are only MoleculeNet and Non-Chirality version MoleculeNet pretrained while QM9 is trained directly?
2. If we calculate a similarity based on the molecular structure using chemical software like RDKit to use as a weight, should we achieve better performance? What do you think?

**Limitations:**

See **Weaknesses** Section.

---

> ### Author Rebuttal · Authors · 2024-08-07
>
> Thank you for your valuable feedback and for recognizing the innovation of our work.
>
> W1: Eq.3 is supposed to be $e^{w_i^+s_i}$, not $w_i^+s_i$. Or am I misunderstanding?
>
> This might be a misunderstanding, Equation 3 is correct as written. This equation is part of our Bayesian augmentation method, designed to create a joint distribution that maintains conjugacy between the priors and posteriors of the weights.
>
> W2: Line 129 contains a duplicated "positive."
>
> We appreciate your careful review and will remove the duplicated.
>
> W3: The authors should explain the function of an augmented random variable ui in the appendix, in case readers are not familiar with the method.
>
> We will add a detailed explanation of the function of the augmented random variable u_i​ in the appendix, providing context and clarity on its role in our method.
>
> Q1: What is the augmentation strategy for QM9 and MoleculeNet? Are only MoleculeNet and Non-Chirality version MoleculeNet pretrained while QM9 is trained directly?
>
> A1: We will add this in the appendix.
>
> For non-chiral MoleculeNet, we apply a combination of node removal and edge removal as augmentation strategies, as shown in Figure 1.
>
> For chiral MoleculeNet and QM9, we adopt two different augmentations, following Uni-Mol:
>
> Atom Masking Augmentation: This augmentation randomly masks a certain percentage of atoms.
>
> Coordinate Masking: This augmentation adds random gaussian noise to atom coordinates.
>
> Also, QM9 has been trained directly without the pretraining on a larger dataset.
>
> Q2: If we calculate a similarity based on the molecular structure using chemical software like RDKit to use as a weight, should we achieve better performance? What do you think?
>
> A2:
> As discussed in our common response to all reviewers, we have conducted experiments comparing different weighting strategies, including similarity-based weights derived from molecular fingerprints based on RDkit.
>
> Our experiments demonstrate that while this similarity-based approach shows some improvement in certain tasks, it does not outperform our proposed method in the majority of the tasks. This suggests that the added complexity and adaptability of our method provides better alignment with the diverse tasks in molecular property prediction.

---

> > ### Comment · Reviewer_Jokw · 2024-08-07
> > **Official Comment by Reviewer Jokw**
> >
> > Thanks for conducting the experiment and your rebuttal; my concern has been resolved, leading me to adjust my rating from 6 to 7.

---

> > > ### Author Response · Authors · 2024-08-09
> > > **Gratitude for Your Review and Enhanced Rating**
> > >
> > > Dear Reviewer,
> > >
> > > Thank you for taking the time. We really appreciate your decision. Your support is invaluable in improving our work.
> > >
> > > Best regards,
> > >
> > > Authors

---

### Official Review · Reviewer_T3Hf · 2024-07-07

**Soundness:** 2
**Presentation:** 3
**Contribution:** 2
**Rating:** 6
**Confidence:** 4

**Summary:**

Commonly used data augmentation methods may produce false positives or negatives in learning molecular representations. This study proposes a novel probability-based contrastive learning framework to tackle false positive and negative pairs in molecular representation learning. Bayesian modeling is used to learn a weight distribution on sample pairs, automatically identifying and mitigating false pairs. This dynamic adjustment enhances representation accuracy. The model is optimized through a stochastic expectation-maximization process that iteratively refines sample weight probabilities and updates model parameters.

**Strengths:**

* Present a good analysis on the impact of false positive and negative pairs
* The proposed probability-weighted CL is novel
* The overall structure of the manuscript is clear and easy to follow (make the fonts in the figures bigger, they are hard to read).
* The experimental results show the average performance. How about variances, an indication of robustness?

**Weaknesses:**

* Why not design the probability distribution based on the similarity score distribution, a simpler approach? The best hyperparameters (Table 6) indicate that the positive and negative pairs have quite different distributions. Maybe a even simpler approach using a similarity thresholding would work.
* What are the appropriate values for a_u and b_u.
* Figure 3 is confusing. Is it used to show that, before learning, some positive samples are highly similar to negative samples, which is no longer the case after learning?
* Recent and better performance on the MoleculeNet datasets should be considered.

[1] Fang, Yin, et al. "Knowledge graph-enhanced molecular contrastive learning with functional prompt." Nature Machine Intelligence 5.5 (2023): 542-553.
[2] Hajiabolhassan, Hossein, et al. "FunQG: Molecular representation learning via quotient graphs." Journal of chemical information and modeling 63.11 (2023): 3275-3287.
[3] Ren, Gao-Peng, Ke-Jun Wu, and Yuchen He. "Enhancing molecular representations via graph transformation layers." Journal of Chemical Information and Modeling 63.9 (2023): 2679-2688.

**Questions:**

* Figure 1, which node(s) and edge(s) are removed? How is similarity calculated? Does augmentation consider structural validity and stability?
* Why do you choose Gamma or Bernoulli distribution over other distribution?
* The authors pointed out that i-MolCLR did something similar, however, claimed that the proposed approach offered a better solution. Why not compare with i-MolCLR in all experiments?
* The paper shows that the best setting is positive pair weights following  a+ =5 and b+ = 1, and negative pairs following a- = 1 and b- = 1. The significant difference between these two distributions is approximately close to the threshold for decision-making. When the distributions are similar, such as positive pair weights following a+ =1 and b+ = 1 while negative pair weights following a- =1 and b- = 1, performance is worse according to the ablation study. Besides, the author should discuss about the choices of au and bu.

**Limitations:**

None.

---

> ### Author Rebuttal · Authors · 2024-08-07
>
> Thank you for your valuable feedback and for recognizing the presentation and novelty of our work.
>
> W1: Why not design the probability distribution based on the similarity score distribution, a simpler approach?
>
> We agree that using a similarity thresholding is a simpler approach. However, as the experiments showed, even if we use fingerprint-based similarity calculation, the issue cannot be completely resolved, and the performance gaps between this simple approach and our methods are in fact still reasonably large. Please refer to our response to all the reviewers for more detailed analysis.
>
> W2: What are the appropriate values for a_u and b_u.
>
> In code, we set a_u=b_u=1, here we conducted an ablation study on the choice of a_u and b_u. Generally speaking, the choice of a_u and b_u will not influence the experiment results to a large margin, the top performance is at a_u=b_u=5.
>
> | a_u          | 1    | 5    | 10   | 1    | 5        | 10   | 1    | 5    | 10   |
> |--------------|------|------|------|------|----------|------|------|------|------|
> | b_u          | 1    | 1    | 1    | 5    | 5        | 5    | 10   | 10   | 10   |
> | Avg ROC_AUC% | 80.4 | 80.2 | 80.1 | 80.6 | **80.7** | 80.2 | 80.3 | 80.1 | 80.5 |
> |              |      |      |      |      |          |      |      |      |      |
>
> W3: Figure 3 is confusing. Is it used to show that, before learning, some positive samples are highly similar to negative samples, which is no longer the case after learning?
>
> Figure 3 is not to illustrate the similarity of positive samples to negative samples before and after learning. Instead, it compares the distributions of similarity scores for positive and negative pairs after pre-training with the original MolCLR loss and the proposed loss.
>
> Left plot (Positive samples): It shows the distribution of similarity scores for positive samples. The proposed method (blue) produces higher similarity scores with a higher mean and lower variance compared to the original MolCLR method (orange). This indicates that the positive samples are more consistently similar to each other with the proposed method.
>
> Right plot (Negative samples): It shows the distribution of similarity scores for negative samples. The proposed method (blue) yields lower mean and lower variance in similarity scores compared to MolCLR. MolCLR shows two peaks (around 1 and 2.7), suggesting that it incorrectly assigns higher similarity to some negative pairs. The proposed method concentrates the negative similarity scores around 1, indicating better separation of negative pairs from positive pairs.
>
> W4 Recent and better performance on the MoleculeNet datasets should be considered.
>
> We appreciate the reviewer's suggestion. We will cite these papers in our related work section. In fact, we can apply our method on top of these methods, we will update the experiments in final version of our paper.
>
> However, we would like to emphasize that direct comparisons with these methods may not be fair due to differences in additional information sources and data splitting methods:
>
> KANO: This method utilizes additional knowledge graph as an information source, which provides a significant advantage. Our method does not use such external information.
>
> LineEvo: This method employs random data splitting, which can yield artificially higher performance.
>
> FunQG: This method uses a molecular graph coarsening framework that significantly reduces the graph's complexity, losing 3D information of each individual atom. Our approach retains the original graph structure.
>
> Here, we present another fairer comparison with a recent baseline Instruct-Bio [1]. The following analysis shows that our method outperforms InstructBio across almost all subtasks in the MoleculeNet classification and regression experiments.
>
> |               |  BBBP        | Tox21        | ToxCast      | SIDER         | ClinTox       | BACE          | ESOL             | FreeSolv        | Lipo             |
> |---------------|--------------|--------------|--------------|---------------|---------------|---------------|------------------|-----------------|------------------|
> | Insturct-bio[1] | 67.4±0.5     | 75.6±0.3     | 65.1±1.5     | 61.5±1.9      | 78.0±0.6      | 78.5±0.8      | 1.771±0.015      | **0.832±0.020** | 0.752±0.017      |
> | Ours          | **76.7±2.0** | **80.1±1.0** | **69.9±2.5** | **64.9± 3.3** | **89.4± 0.1** | **88.2± 1.3** | **0.664± 0.024** | 1.358± 0.452    | **0.590± 0.003** |
> |               |              |              |              |               |               |               |                  |                 |                  |
>
> [1] Wu F, Qin H, Li S, et al. Instructbio: A large-scale semi-supervised learning paradigm for biochemical problems[J]. arXiv preprint arXiv:2304.03906, 2023.

---

> ### Author Response · Authors · 2024-08-07
> **Rebuttal Continued**
>
> Questions:
>
> Q1: Figure 1, which node(s) and edge(s) are removed? How is similarity calculated? Does augmentation consider structural validity and stability?
>
> [A1]  Sorry for making the confusion, we will add this to explain in our manuscript.
>
> In Figure 1, the nodes marked with red dots and the edges marked with red lines are the ones removed. When we “remove” a node, we do not entirely eliminate it from the graph. Instead, we substitute it with a special [MSK] node that does not correspond to a specific element type(like C or O). This approach is employed to avoid any changes in the topology that would occur due to node removal. Structural validity and stability is also preserved.
>
> The similarity is calculated based on the representations from our trained encoders. The reason for not using fingerprint-based similarity is that such methods are specifically designed for unmodified molecules. The introduction of [MSK] nodes leads to discrepancies when using fingerprint-based similarities, as these traditional methods cannot accommodate the presence of masked nodes effectively. Therefore, we rely on our model's learned representations, which are robust to such augmentations, to measure similarity accurately.
>
> Q2: Why do you choose Gamma or Bernoulli distribution over other distribution?
>
> A2: We use the Gamma prior because it naturally lends itself to conjugacy in the posterior, which significantly eases the posterior sampling procedure. Also, it is known for its flexibility in shape and scale to model positive continuous variables, which is suitable for sample weights in our setting. Additionally, we compared it with Bernoulli prior which uses binary sample weights in Table 1. Our results show that Gamma prior offers superior robustness and accuracy. This advantage is likely due to the broader search space available to continuous weights, with binary weights being a subset of this continuous space.
>
> Q3: The authors pointed out that i-MolCLR did something similar, however, claimed that the proposed approach offered a better solution. Why not compare with i-MolCLR in all experiments?
>
> A3: We appreciate your suggestion. Please refer to the common reply to all reviewers for experiment using fingerprint-based similarity on Chiral MoleculeNet dataset (that is exactly the method proposed in I-MolCLR) and tabel 3 for experiment on non-Chiral MoleculeNet dataset.
>
> We are currently working on extending our experiments to include QM9 dataset, and we will incorporate these results in a future version of the paper.
>
> Q4: The paper shows that the best setting is positive pair weights following a+ =5 and b+ = 1, and negative pairs following a- = 1 and b- = 1. The significant difference between these two distributions is approximately close to the threshold for decision-making. When the distributions are similar, such as positive pair weights following a+ =1 and b+ = 1 while negative pair weights following a- =1 and b- = 1, performance is worse according to the ablation study. Besides, the author should discuss about the choices of au and bu.
>
> A4: We agree that the choice of hyperparameters in the prior distribution plays a crucial role in the performance of our method. The significant difference between the distributions for positive and negative pairs reflect the inherent differences in their nature. This also suggests that the model requires a more pronounced distinction between these pairs to effectively learn useful representations. Experiments show that the best performance is achieved when au=bu=5, please refer to our answer on W2 for experiment details.

---

> ### Author Response · Authors · 2024-08-09
> **Follow-up on Rebuttal: Request for Further Review and Feedback**
>
> Dear Reviewer T3Hf,
>
> Thank you for your thorough feedback on our work. We have carefully addressed each of your comments and provided additional analyses to clarify the points you raised. We believe our detailed rebuttal directly addresses your concerns and strengthens the validity of our approach. We kindly encourage you to review our responses at your earliest convenience, as your insights are invaluable to improving our manuscript.
>
> We appreciate your time and look forward to any further comments you might have.
>
> Best regards,
>
> Authors

---

> > ### Comment · Reviewer_T3Hf · 2024-08-09
> > **I moved my decision from 5 to 6.**
> >
> > The responses address some of my concerns. It will be interesting to see additional experimental results.

---

### Author Rebuttal · Authors · 2024-08-07

We thank the reviewers for their valuable comments. We are happy that the reviewers find our work innovative and promising in general. We notice there are also some questions and concerns from several perspectives. A common issue raised by the reviewers is the comparison against directly using similarity scores threshold to decide weight distributions, which we will address below. For other specific questions raised by each reviewer, we will post our responses separately. We also have revised the manuscript by incorporating some extra results and explanations based on the reviews. We will incorporate more detailed revisions into the camera-ready version according to responses and further discussions. We hope the reviewers can read our response and reconsider your decision if necessary. Thanks again for helping us to make this work better.

Q: Why not design the probability distribution based on the similarity score distribution?

We appreciate the reviewers’ insightful suggestion to use a simpler approach based on similarity scores. To thoroughly investigate this, we designed ablation experiments using the chiral version of MoleculeNet classification tasks and compared three different methods:

1. Bayesian Inference (Original Method): In this approach, we calculate the weights using Bayesian inference, as described in our paper.

2. Fingerprint-based Similarity: This method calculates the weights based on the similarity scores derived from molecular fingerprints, similar to the approach used in I-MolCLR.

3. Encoder-based Similarity: Here, we first extract features of data pairs using encoders and then calculate their similarity scores. These scores are then regularized to the [0,1] range.

For methods 2 and 3, we compute the weights using the following formulas:

$w_i^- = 1 - \lambda \times Sim(x_i, x_k)$,

$w_i^+ = \lambda \times Sim(x_i, x_k)$

In our experiments, we set $\(\lambda = 1\)$.

|  MoleculeNet Experiments |  BBBP     | Tox21     | ToxCast   | SIDER     | ClinTox   | BACE      | MUV       | HIV       | PCBA      |
|--------------------------|-----------|-----------|-----------|-----------|-----------|-----------|-----------|-----------|-----------|
| Bayesian Inference       | 76.7± 2.0 | **80.1± 1.0** | **69.9± 2.5** | **64.9± 3.3** | 89.4± 0.1 | **88.2± 1.3** | **82.9± 3.1** | **83.0± 1.7** | 82.9± 0.5 |
| Fingerprint Similarity   | **77.9**      | 80.0      | 68.9      | 64.9      | 87.6      | 83.6      | 78.7      | 79.8      | **83.8**      |
| Encoder-based Similarity | 75.2      | 79.6      | 67.8      | 58.5      | **90.4**      | 82.8      | 80.5      | 81.0      | 74.5      |

From these results, we observe the following:

1. Fingerprint-based Similarity: While this method showed improvements in 2 out of 9 tasks compared to our original method, but it did not perform as well overall. This indicates that while similarity-based methods are simple, they may not be flexible enough to fully capture the complexities of molecular representations required for robust performance across diverse tasks.

2. Encoder-based Similarity: This approach performed worse than both the Bayesian inference method and the fingerprint-based similarity approach, further suggesting that using a direct similarity-based method does not necessarily yield better results.

These findings suggest that while simpler methods like similarity thresholding may work in some cases, they do not outperform our proposed Bayesian inference method which is designed to dynamically adapt and provide better alignment of positive and negative pairs of molecular representation learning. Thus, our Bayesian inference-based approach is essential for achieving state-of-the-art performance across various molecular property prediction benchmarks.

We hope this detailed comparison addresses the reviewer’s concerns and provides clarity on the advantages of our proposed method.

---

### Decision · Program_Chairs · 2024-09-25

**Decision:**

Accept (poster)

**Comment:**

This paper proposes a contrastive learning framework for molecular representation learning. The main idea is to remove the "false" pairs of data augmentation from severely modifying the graph structure. The authors propose a Bayesian learning framework to solve this problem.

Most of the reviewers strongly support the paper. One reviewer raised concerns on (1) justification behind the Gamma prior, (2) Figure copyrights, and (3) missing discussion of related works. However, after carefully reading through the author rebuttal, I believe the author rebuttal is persuasive enough to alleviate the concerns.

Therefore, I recommend acceptance for this work.